New observations on test architecture and construction of Jullienella foetida Schlumberger, 1890, the largest shallow-water agglutinated foraminifer in modern oceans

Langer Martin R. 1 martin.langer@uni-bonn.de
http://orcid.org/0000-0002-7999-0486 Weinmann Anna E. 2
Makled Walid A. 3
Könen Janine 1
Gooday Andrew J. 4 5
1 Institute of Geoscience, Paleontology, Rheinische Friedrich-Wilhelms Universität Bonn , Bonn , Germany
2 Geological-Paleontological Department, Natural History Museum Vienna , Vienna , Austria
3 Exploration Department, Egyptian Petroleum Research Institute (EPRI) , Cairo , Egypt
4 National Oceanography Centre , Southampton , United Kingdom
5 Life Sciences Department, Natural History Museum , London , United Kingdom
De Baets Kenneth
Electronic publication date: 2022 Feb 15
Publication date: 2022
Volume: 10
Electronic Location ID: e12884
Received 2021 Nov 3; Accepted 2022 Jan 13
Copyright: © 2022 Langer et al.
Copyright year: 2022
Copyright holder: Langer et al.
License: This is an open access article distributed under the terms of the Creative Commons Attribution License, which permits unrestricted use, distribution, reproduction and adaptation in any medium and for any purpose provided that it is properly attributed. For attribution, the original author(s), title, publication source (PeerJ) and either DOI or URL of the article must be cited.
License URL: https://creativecommons.org/licenses/by/4.0/

Keywords: Foraminifera, Micro-CT, West Africa, Biomass, Test construction, X-ray, Morphology, Upwelling

Funding: German Science Foundation LA 884/14-1 This work was supported by the German Science Foundation (No. LA 884/14-1). The funders had no role in study design, data collection and analysis, decision to publish, or preparation of the manuscript.

==============================
We present new observations on Jullienella foetida Schlumberger, 1890, a giant agglutinated foraminifer with a leaf- or fan-like test reaching a maximum dimension of 14 cm, that is common on some parts of the west African continental shelf. The test wall comprises a smooth, outer veneer of small (<10 µm) mineral grains that overlies the much thicker inner layer, which has a porous structure and is composed of grains measuring several hundreds of microns in size. Micro-CT scans suggest that much of the test interior is filled with cytoplasm, while X-ray micrographs reveal an elaborate system of radiating internal partitions that probably serve to channel cytoplasmic flow and strengthen the test. Jullienella foetida resembles some xenophyophores (giant deep-sea foraminifera) in terms of test size and morphology, but lacks their distinctive internal organization; the similarities are therefore likely to be convergent. Based on micro-CT scan data, we calculated an individual cytoplasmic biomass of 3.65 mg wet weight for one specimen. When combined with literature records of seafloor coverage, this yielded an estimate of >7.0 g wet weight m−2 for the seafloor biomass of J. foetida in areas where it is particularly abundant. The relatively restricted distribution of this species off the north-west African coast at depths above 100 m is probably related to the elevated, upwelling-related surface productivity along this margin, which provides enough food to sustain this high biomass. This remarkable species appears to play an important, perhaps keystone, role in benthic ecosystems where it is abundant, providing the only common hard substrate on which sessile organisms can settle.

Introduction

In 1890, Schlumberger described a new and gigantic agglutinated foraminifer from the western coast of Africa (Liberia) and named it Jullienella foetida after its collector, the French bryozoan specialist Jules Jullien (Schlumberger, 1890). When first discovered, during a French expedition off Liberia in front of “Poor River” at 12.6 m water depth (Wedabo Beach), Jullien noted that the specimens exuded a particularly “foul-smelling odour”, leading Schlumberger to name it foetida (from lat. foetidus meaning fetid, foetid or malodorous). The species was initially considered to be a bryozoan, but Schlumberger recognized its true character and correctly described it as a single-chambered (monothalamous) agglutinated foraminifer with a large, flat or slightly undulating plate-like test, leaf-like, or fan-like in overall shape and with the chamber interior subdivided by longitudinal partitions (Schlumberger, 1890).

Jullienella foetida is currently placed within the Schizamminidae, a family established by Nørvang (1961) for species of large and somewhat bizarre monothalamous foraminifera that includes the genus Schizammina Heron-Allen & Earland, 1929, in addition to Jullienella Schlumberger, 1890. This family belongs to the class Monothalamea (‘monothalamids’), a paraphyletic group of single-chambered foraminifera that encompasses the orders Allogromiida and Astrorhizida, and includes freshwater as well as marine species (Pawlowski, Holzmann & Tyszka, 2013). The monothalamids are subject to ongoing genetically-based revisions and species are currently grouped into a series of clades (Voltski & Pawlowski, 2015; Gooday et al., 2020). Unfortunately, there are no genetic data for any species of the Schizamminidae and the relationship of these unusual foraminifera to other monothalamids is therefore unclear.

Since it was first described by Schlumberger (1890), J. foetida has been widely reported from depths of 14 to 89 m across the West African continental shelf from Western Sahara to Ghana (Fig. 1) (Longhurst, 1958; Buchanan, 1958, 1960; Nørvang, 1961; Le Calvez, 1963, 1972; Manning & Holthuis, 1981). It occurs on fine sandy and muddy substrates at densities of up to 200 individuals per m2 and covering up to 10% of the sandy seafloor (Le Loeuff & Intes, 1968; Thiel, 1982; Tendal & Thiel, 2003). Schlumberger (1890) reported that the largest specimens from off Liberia were 6 cm in maximum dimension, and more recent records suggest that it can reach more than twice this size. In situ images of J. foetida have shown the thin, plate-like test lying horizontally on the sediment surface with only the lower side partially buried (Thiel, 1982; Tendal & Thiel, 2003). These large agglutinated structures often constitute the only available hard substrate on which sessile organisms can settle (Cook, 1968, 1985).

Figure 1 Distribution of Jullienella foetida.

Map showing all known localities where Jullienella foetida has been recorded. The unpublished record from off Ghana is based on a sample in the collections of the National Oceanography Centre, Southampton, of uncertain provenance. The label in the bottle reads ‘Plant material. Agazziz Trawl No 3. 2-5-51 (i.e., 2nd of May 1951). Gold Coast. R. Barrindale’.

Its great size, abundance and ecological importance make J. foetida an important species in some continental shelf ecosystems of the west African shelf. The main goals of this study were to (1) explore its external and internal test characteristics using a suite of non-destructive methods, namely light microscopy, SEM, X-ray and high-resolution X-ray micro-computed tomography (micro-CT), and (2) provide a first-order estimate of its possible contribution to sea floor biomass. In addition, we compare this remarkable shallow-water species to another group of giant monothalamid foraminifera, the deep-sea xenophyophores, to which it displays a striking morphological similarity.

Materials and Methods

The new material of Jullienella foetida originates from sediment samples collected using a box corer in 1971 during Meteor Cruise M25 at a water depth of 68 m (sample station # 74/1; Seibold, 1971, 1972). The sample site is located off the coast of Mauritania, north of the capital of Nouakchott at 18°52′N and 16°31′W (Fig. 1). Upon collection, foraminifera were picked from the sediment surface and dried at room temperature. A total of 12 tests was examined (Fig. 2). Images were taken using light microscopy and Scanning Electron Microscopy (SEM, CamScan MV 2300, Vegascan) and arranged into plates using Adobe Photoshop CS6. X-ray pictures were obtained using a Radifluor 360 generator (Philips Electronics). All are stored in the micropaleontology collection at the Institute of Geoscience, University of Bonn (LA-2021-Jf-1-14). Some additional observations were made on specimens, stored at the National Oceanography Centre, Southampton. The label indicates that they were collected on 2 May 1951 with an Agassiz trawl by R. Bassindale off the coast of Ghana. No further information is available about the provenance of this material (included as ‘Unpublished record’ in Fig. 1).

Figure 2 Jullienella foetida.

Jullienella foetida; light photographs and corresponding X-ray photographs of 10 specimens. The radiating linear structures in the X-ray images are interpreted as internal partitions. In some cases, these features are strongly developed along their entire length, but in others they resemble dashed lines, with prominent sections separated by gaps where they are weakly developed or absent.

To investigate the internal structure further, two individuals of J. foetida (Figs. 2A and 2F) were scanned using the micro-CT scanner v|tome|xs 240 kV (GE Sensing & Inspection Technologies GmbH phoenix|x-ray) at the Institute of Geosciences, University of Bonn. During the scans, a total of 1,000 X-ray projections was collected through a 360° rotation of the sample. The specimens were scanned dry at 120 kV and 120 µA (voxel size 0.016 mm). The micro-CT scanner is equipped with a detector panel that produces isotropic voxels (single-size image 2,024 × 2,024 pixels) and a maximum resolution (voxel size) of 1 μm. For the present study, the size of the specimens meant that they had to be positioned some distance from the detector. This resulted in a reduced resolution (16 μm voxel size), so that finer structures could not be seen. For all scans, the same shutter speed of 200 ms per capture was used. This generated a stack of grayscale JPEG slice images that were imported into the visualization and analysis program Avizo light 9.2 (ThermoFisher Scientific, Waltham, MA, USA) for the segmentation of individual architectural elements based on grayscale values (relative X-ray absorption). 3D-reconstructions of the test and volumetric calculations for the test and chamber lumina were then generated, again using Avizo. The raw CT scans and reconstructed 3D models are available for viewing and download on MorphoSource (http://www.morphosource.org/) in Project 000393778.

Micro-CT imaging, which has been applied only occasionally to agglutinated foraminifera, was used here to visualize the distribution of the cytoplasm and the test simultaneously. To calculate the area occupied by remnants of dried cytoplasm in vertical and horizontal micro-CT stack images, grey-scale images were analysed using ImageJ software (Rasband, 1997–2018). The resulting image analysis provides novel information about the relationship between the cytoplasm and the test.

Results

The following descriptions are based mainly on specimens from the Mauritanian margin, but with some additional information derived from the Ghanaian material.

Overall test morphology

Our specimens of Jullienella foetida have hard, rigid, leaf-like, fan-like, and plate-like agglutinated tests (Fig. 2), up to ~3 cm in size. The surface is wrinkled, often gently undulating and interrupted by more or less distinct arcuate or crescentic ridges (Figs. 2C, 2G), spaced at intervals of about 1.5 mm. Total test thickness ranges between 800 µm and 1.2 mm (n = 8), with lowest values in smaller individuals and within the slightly depressed areas between ridges. The wall thickness does not vary significantly during growth stages and remains almost constant throughout the test. In one specimen (Fig. 3A), the plate-like test has overgrown three finger-like projections of what appears to have been an earlier outer margin. Possibly, the growth of this specimen had been interrupted and then redirected as a result of some damage or trauma.

Figure 3 SEM electron micrographs of Jullienella foetida.

Jullienella foetida; scanning electron micrographs. (A) Complete specimen. (B) Test surface showing smooth, fine-grained outer layer with larger grains projecting through it from the underlying wall. (C) Detail of area enclosed by rectangle in figure (A) showing area where the wall has been removed to show internal features and remnant of dried cytoplasm (smaller rectangle). (D) Detail of area indicated by the larger rectangle in figure (C) showing inner surface of test wall with complex pattern of pits and upstanding areas. (E) Detail of area indicated by smaller rectangle in figure (C) showing surface of cytoplasmic remnant. (F) Detail of area indicated by rectangle in figure (E) showing surface of cytoplasm with diatoms.

The proximal part of the test often consists of a flat stem of varying width and length, with an open end and no sign of an initial structure. Two specimens show a bifurcation of the stem (Figs. 2B, 2E), possibly the remains of a link between two lobes of the test (see also Nørvang, 1961), rather than the initial part. Towards the distal end of the test, flattened tubular processes, usually fairly short, extend from the margin, sometimes branching dichotomously (Figs. 2A, 2F, 2I; Figs. 4A, 4B: Fig. 5A; Fig. 6B). The Mauritanian and Ghanaian specimens display a variety of marginal features, including arcuate sections that are either uninterrupted or divided into shorter sections by recesses of the edge, more discrete, flattened, sometimes branching processes, and tubular processes that may arise from the surface of a plate and are orientated in different directions (Figs. 7A, 7C, 7F). These marginal features typically have light-coloured extremities and are probably sites of active test growth from which pseudopodia are deployed. In many cases, the ends are blocked by a plug of large grains (Figs. 7B, 7D, 7E). Open extremities usually lack the lighter colour, and greyish cytoplasm may be visible within them (Fig. 7G).

Figure 4 Jullienella foetida; micro-CT scans.

Jullienella foetida; micro-CT scans of the two specimens shown in Figs. 2A, 2F. (A), (C) Test surface; note the protruding grains. (B), (D) Test lumen showing the interface between the test wall and the inner cavity. In effect, this is a view of the inner surface of the wall in reverse. The interface is covered with small-scale irregularities reflecting the labyrinthic nature of the test wall. Note that the internal partitions in (B), (D), indicated by open spaces, are developed intermittently, particularly in (B). MorphoSource ARK identifiers (A) ark:/87602/m4/393825; (B): ark:/87602/m4/393822; (C): ark:/87602/m4/393850; (D): ark:/87602/m4/393854.

Figure 5 Scanning electron micrographs of Jullienella foetida.

Jullienella foetida; scanning electron micrographs. (A) Complete specimen. (B–D) Progressively closer views of a broken edge with the coarsely agglutinated test wall overlain by a very thin, fine-grained surface veneer; the test lumen is interrupted by internal agglutinated grains that form, either cross-sections of partitions or more isolated columnar structures. (E) Detail of broken test wall showing large agglutinated grains with intervening spaces filled by fine-grained mortar. (F) Detail of fine-grained mortar.

Figure 6 Micro-CT images of Jullienella specimen with coloured cytoplasm.

Greyscale micro-CT images of the Jullienella specimen illustrated in Figs. 2A, 2A’ and 4A, 4B with two density components. (A) Cross section. (B) Section in the plane of the test. The agglutinated test wall (=aw) and internal test partitions (=itp) are well defined as dense and bright white in greyscale scan images. The cytoplasm (=cy) occurs as low-density, material (coloured yellow) and is patchily distributed throughout the test. See also Fig. S1. MorphoSource ARK identifiers (A) ark:/87602/m4/393828; (B) ark:/87602/m4/393831.

Figure 7 Light microscope images of “apertural” features.

Light microscope images of “apertural” features along the margin of Jullienella foetida. (A–H) Specimens from off Ghana. (A) Tubular and wider, flattened processes at the peripheral margin of a test. (B), (C) Details of the terminal surfaces of marginal processes with plugs of large grains. (D) Second specimen with tubular processes that arise from the surface of a plate in different directions, and wider, flattened marginal processes. (E) Blunt ends of tubular processes with plugs of large grains. (F) Arcuate and uninterrupted peripheral margin. Note that the marginal area comprises an inner orange-coloured zone and a narrower, pale edge; it presumably indicates recent growth. (G), (H) Detail showing a damaged section of the margin, exposing greyish cytoplasm. (I) Specimen fragment from off Mauritania partly broken open to show cytoplasm. Note the attached epifauna in (A) and (D).

Test structure

Examination of Jullienella foetida tests by SEM revealed an external veneer, comprising small (typically 5–10 µm) angular mineral particles (Figs. 5C, 5D). The layer is very thin (~20 µm) with a smooth outer surface, but is interrupted by numerous shallow bumps where the much larger underlying grains protrude through it (Fig. 3B). This arrangement, and the transparency of the protruding grains, creates a finely speckled appearance when the wall is viewed at high magnifications under a light microscope. The underlying wall is much thicker (250–350 µm) and composed of subrounded grains, measuring several 100s of microns in size (Figs. 5C, 5D). It has a very irregular inner surface with deep pits that communicate with open spaces within the wall, creating a porous, labyrinthic structure (Fig. 3D). This is reflected in the micro-CT scans, which show a reversed image of the inner surface of the wall in which the pits appear as tiny projections (Figs. 4B, 4D). To some extent the spaces between the grains are occupied by fine particles similar to those in the outer veneer (Figs. 5E, 5F). The internal partitions are extensions of the inner layer of the wall and have a similar structure.

Internal structure

The volume of the test and of the internal cavity could be derived from micro-CT data. In two specimens, the agglutinated test alone occupied 67% and 73% (mean 70%) and the test cavity 33% and 27% (mean 30%) of the total volume. Scanning electron, X-ray and micro-CT images show that the test cavity of Jullienella foetida is subdivided by a series of discontinuous radiating walls (internal partitions) that have no external expression on the outer surface of the test (Figs. 2, 4, 5). These partitions are aligned almost in parallel and are spaced at regular distances, subdividing the test lumen into elongated sections. Their radial arrangement reflects the fan-shaped and leaf-like growth form of the large agglutinated test. As the lateral fanning-out of the test increases with growth, new partitions are added (Figs. 2D', 2J'). Almost all these internal walls are discontinuous, with interruptions often occurring at approximately the same growth stage, allowing efficient protoplasmic communication in both longitudinal and lateral directions. From an architectural point of view, the longitudinal arrangement of the intermittent partitions may serve to strengthen the test and prevent the otherwise unsupported “roof” from collapsing, in addition to channelling cytoplasmic streaming. However, they are not equally developed in all specimens. In some (Figs. 2D', 2I', 2J'), they appear consistently strong in the X-ray images, but in others they are weaker and more intermittent (Figs. 2A', 2C', 2G').

Cytoplasm

Along broken edges of the test, remnants of dark brown cytoplasm are present within the lumen (Fig. 5D) or attached to the inner surface of the wall (Figs. 3E, 3F). This material contains a number of diatoms (Fig. 3F). What we interpret as cytoplasm is also visible in micro-CT scans, where it stands out as a light-grey, low-density component within the test lumen. In the CT-scan slices shown in Figs. 6A, 6B, it is present within the central area as well as the finger-like projections and marginal openings, but is patchily distributed. Grey-scale image analysis using ImageJ showed that the dried cytoplasm occupies ~19.2% of the test area in cross section (Fig. 6A), and ~36.4% in vertical sections (Fig. 6B). However, different horizontal CT-scan slices of the same specimen (Fig. S1) reveal the presence of cytoplasm in other parts of the test lumen. This suggests that cytoplasm is more widely distributed within the test lumen than is apparent in Fig. 6B, although shrinkage during drying will have created gaps.

Discussion

Comparison with previous observations

The largest specimens of Jullienella foetida documented in the literature were observed at ~60 m depth off the coast of Mauritania (station 192, Meteor expedition 44; Tendal & Thiel, 2003). Here, they reached a maximum dimension of 14 cm and included a range of morphological types, including thin, leaf-, fan- and kidney-shaped forms, most of them more or less flat. Other published illustrations show the leaf-like growth pattern extending from a central juvenile stage in opposite directions to create a dumbbell-shaped test, or in one direction to form a subcircular feature (Schlumberger, 1890; Buchanan, 1958; Longhurst, 1958; Nørvang, 1961; Tendal & Thiel, 2003; see also our Figs. 2A, 2G). Finger-like tubular processes extending from the test margin, are characteristic of the species (Schlumberger, 1890; Buchanan, 1960; Nørvang, 1961). Altenbach et al. (2003) described these features as extending laterally or at an angle of 90° into the seafloor, occasionally branching at some distance from the main part of the test (Tendal & Thiel, 2003). Some of our Ghanaian specimens also incorporate tubular processes that project at various angles, sometimes from main plane of the test (Fig. 7D; Gooday, unpublished observations). The lighter colour of test extremities, where growth is presumably occurring, was mentioned by Nørvang (1961) who, as in the present study, observed that they are often plugged by large mineral grains (Figs. 7B, 7D, 7E). Nørvang also notes that the grains are embedded in protoplasm, so these ‘plugged’ extremities may still be able to function as the apertures for the pseudopodia.

Our Mauritanian specimens of J. foetida have maximum dimensions of only ~3 cm and are therefore much smaller than many of those illustrated in the literature, including the specimens of Tendal & Thiel (2003), referred to above, which were also from the Mauritanian margin. None appears to be intact, and indeed, complete specimens of this species have rarely been recovered (Buchanan, 1958). They are all plate-like (Fig. 2), and one incorporates a single open space (Fig. 3A), but there is no tendency to form a reticulated structure, as seen in some examples illustrated by Nørvang (1961) and Tendal & Thiel (2003). The marginal processes are also generally less well developed, but in other respects, our specimens resemble published illustration of J. foetida and we have no doubt that they belong to this large and distinctive species. Those from the Ghanaian margin are often larger, up to 4 to 5 cm in maximum extent and display a variety of well-developed marginal structures (Figs. 7B, 7D).

The discontinuous, fine-grained surface veneer with protruding large grains is reminiscent of the pattern of agglutination seen in Astrammina rara Rhumbler, 1931 (Bowser & Bernhard, 1993), although the surface layer in J. foetida is more distinct and the protruding grains occupy a smaller area than in A. rara. In common with many other agglutinated foraminiferal species (Heron-Allen, 1915; Lipps, 1973; Armynot du Chatelet, Recourt & Chopin, 2008; Makled & Langer, 2009), J. foetida appears able to select particular kinds of grains according to both size and composition (Nørvang, 1961; but see Buchanan, 1960). Schlumberger (1890) described a brown chitinous cement (‘une matière chitineuse brune’) binding together the larger grains. Loeblich & Tappan (1955, 1987) found no evidence of chitin, but instead mention large amounts of ferruginous cement, insoluble in hydrochloric acid. According to Berthois & Le Calvez (1966), the test of J. foetida is formed from quartz grains with a siliceous cement (‘ciment siliceux’) secreted by the organism. However, no further details of the cement are given and it is unclear what the authors were referring to. Loeblich & Tappan (1989) conclude that there is no evidence for siliceous cement in any agglutinated foraminifera. The cement reported in these earlier studies may refer to the fine-grained particles that occupy the space between the larger grains. We assume that some kind of organic cement, similar to that found in other agglutinated foraminifera (Bender, 1989, 1995; Loeblich & Tappan, 1989; Kaminski, 2004), is present, although we did not observe any obvious examples in our SEM micrographs (Figs. 5E, 5F). There is also no evidence in our material of a ‘rather thick layer of tectin’ coating the test surface (p. 196 in Nørvang, 1961), or of an inner organic lining, although an inner lining is present in the bathyal species J. zealandica (Hayward & Gordon, 1984).

The generic diagnosis of Jullienella given by Loeblich & Tappan (1955) states that the ‘interior surface (of the test) has many large pores, which apparently connect with much more restricted openings at the surface’. Small, round, external openings are also mentioned by Loeblich & Tappan (1987). However, in agreement with the observations of Buchanan (1960) and Nørvang (1961), we found no trace of any openings on the surfaces of our specimens.

The possible contribution of Jullienella foetida to seafloor biomass

SEM and light microscope observations of J. foetida demonstrate the presence of cytoplasm within at least some of the tests. In SEM images, the dried cytoplasm appears granular and contains scattered diatoms (Fig. 3F). Our micro-CT data provide further information, suggesting that the cytoplasm is distributed throughout the test. These scans allow us to make rough estimates of the individual biomass of the specimen illustrated in Figs. 4A, 4B. The absolute volumes of the test and lumen were 190.2 mm3 and 71.56 mm3, respectively. If we assume that 50% of the lumen was occupied by cytoplasm (a very conservative estimate given that patches of cytoplasm were present in most parts of the test; see Fig. 6B and Fig. S1), and that the density of the cytoplasm is 1.02 g ml−1 (= 0.102 mg mm−3), then the individual biomass of this specimen would be 3.65 mg wet weight. Gooday et al. (2018) report cytoplasm (‘granellare’) volumes for 4 abyssal Pacific xenophyophore specimens varying from 9.45 mm3 in Galatheammina sp. to 72.6 mm3 in their specimen 1 of Psammina aff. limbata. Xenophyophore cytoplasm is packed with barite crystals. If we assume that these occupied 50% of the granellare volume in these xenophyophores, then their individual biomass values were between 0.48 to 3.70 mg wet weight, respectively. According to these calculations, our scanned J. foetida specimen had a biomass comparable to that of a slightly larger xenophyophore (the maximum dimension of P. aff. limbata specimen 1 was ~3.5 cm compared to ~2.2 cm for J. foetida), and greater than that of three other xenophyophore specimens.

If similar assumptions are applied to the much larger specimens of J. foetida photographed by Tendal & Thiel (2003) at 17° N off Mauritania, then their individual biomass would be much greater than the estimate for our specimen. Thiel (1982) estimated that Jullienella covered up to about 10% of the seafloor area. Based on these data, and assuming that the average test thickness is 1 mm, the cytoplasm occupies 50% of the lumen, and again that the wet weight of 1 ml of cytoplasm is 1.02 g (Levin & Gooday, 1992, Gooday et al., 2018), we calculate the maximum possible seafloor biomass of J. foetida in Thiel (1982) study area to be 15.3 g wet weight m−2. This estimate would be less if, as seems likely, a proportion of the specimens was dead. For the sake of argument, we will again assume that this proportion is 50%, which would reduce the seafloor biomass to 7.65 g wet weight m−2.

We emphasise that these estimates are based on very limited data and involve several major assumptions, particularly regarding the extrapolation from our study to that of Thiel (1982). The actual figures should therefore not be taken too seriously. However, they are probably the right order of magnitude and give some indication of the contribution that J. foetida could make to seafloor biomass on parts of the NW African shelf (Fig. 1) where it is abundant. The value of 7.65 g wet weight m−2 is comparable to maximum foraminiferal biomass estimates, in most cases derived from whole assemblages of smaller species, from different settings (Murray, 2006). For shelf seas around Europe and North America it is higher than almost all of those (maximum 2.99 g m−2; in one case 16.3 g m−2) compiled by Murray & Alve (2000).

Korsun (2002) concludes that at shelf and upper bathyal depths in parts of the Eurasian Arctic, foraminiferal biomass may be dominated by large agglutinated species. Our estimate is higher than that for the St. Anna Trough in the Kara Sea (0.06–1.7 g m−2), where biomass in the >500-µm sieve fraction was dominated by Reophax pilulifer (Korsun et al., 1998). However, an earlier Russian study cited by Korsun et al. (1998) gives values (1 to 10 g m−2) that are comparable to ours for large astrorhiziid foraminifera (Hyperammina subnodosa, Rhabdammina abyssorum, Pelosina variabilis) in the Barents Sea. At a 230-m-deep site in the Barents Sea, Kuznetsov (1996) recorded a biomass of 6.2 g m−2 for Hormosina globulifera (although his illustration shows a unilocular test resembling Saccammina sphaerica). We therefore believe that our estimates provide plausible maximum values for the seafloor biomass of J. foetida.

Comparison with xenophyophores

In terms of test morphology, Jullienella foetida resembles some xenophyophores, a group of large monothalamous foraminifera (suborder Xenophyophoroidea) that are common in the deep sea below about 550 m depth (Tendal, 1972, 1996; Gooday et al., 2017). These similarities were first noticed by Goës (1892) who considered that J. foetida ‘has much in common’ with Neusina agassizi Goës, 1892, a species he described from the tropical eastern Pacific that is synonymous with the xenophyophore Stannophyllum zonarium Haeckel, 1889. He concluded that the two species ‘stand much isolated’ from other agglutinated foraminifera and ‘justly claim to be placed in a family by themselves’. Later, Cushman (1927) established the family Neusinidae to accommodate Neusina Goës, 1892 and Botellina Carpenter, Jeffreys & Thompson, 1870, to which he later added Jullienella, and Schizammina (Cushman, 1948), apparently unaware of the synonymy between N. agassizi and S. zonarium. Jullienella is in fact quite different from Stannophyllum, which has a soft, flexible test ramified by fine proteinaceous fibres (Tendal, 1972; Gooday et al., 2020). As pointed out by Schulze (1907), these fibres (‘dünne Chitinfäden’) are not present in Jullienella.

There is a closer morphological similarity between J. foetida and some plate-like species of Psammina, such as P. zonaria Tendal, 1994, in which a proximal tube widens to become flat and plate-like (Tendal, 1994). The arrangement of cytoplasmic strands in the fan-shaped P. aff. limbata (Gooday et al., 2018) is reminiscent of the system of partitions in J. foetida, and the corresponding shape of the cell body. Nazareammina tenera Gooday, Aranda da Silva & Pawlowski, 2011 is another Jullienella-like xenophyophore. Photographs of this abyssal species taken on the surface of a box core resemble in situ images of J. foetida (compare Fig. 12A of Gooday, Aranda da Silva & Pawlowski, 2011, with photographs in Tendal & Thiel, 2003). Like some specimens of J. foetida illustrated in Plate VIII, Figs. 1, 2, 5 of Nørvang (1961), N. tenera also has a tendency for the plate-like test to break into bar-like elements that may form a reticulated structure. Despite these morphological similarities, all xenophyophores have a distinctive internal organization, comprising light-coloured strands of cytoplasm enclosed within an organic tube (‘granellare’) and dark accumulations of waste pellets (stercomata), that distinguish them from J. foetida. These features are often immediately obvious when a xenophyophore test is broken open (Gooday et al., 2018) but have never been reported in J. foetida. The similarities in test morphology between these two taxa are very likely to be convergent. However, in the absence of genetic data for J. foetida, a phylogenetic relationship between them cannot be entirely ruled out.

We are somewhat less confident about Jullienella zealandica Hayward & Gordon, 1984. Some specimens of this species illustrated by Hayward & Gordon (1984) are remarkably similar in their overall external test morphology to Psammina zonaria. It is important to note that J. zealandica lives at 950 to 1,400 m, well within the known depth range of xenophyophores (Tendal, 1972, 1989, 1996) but much deeper than other members of the Schizamminidae. However, the internal test structure is apparently rather different, being subdivided by transverse partitions in P. zonarium but undivided in J. zealandica. An examination of the cellular organization of this species would be helpful in determining whether or not it is a true Jullienella species.

Distribution

Since Schlumberger’s original description from off Liberia (Schlumberger, 1890), Jullienella foetida has been found to be widespread along the western coast of Africa, including between Western Sahara to Ghana, Mauritania, Senegal, Gambia, French Guinea, Sierra Leone, Liberia, Ghana, and Côte d’Ivoire (Fig. 1). Our material adds an additional record from off Mauritania. The species occurs across this range in fairly shallow waters on sandy sediment at depths between 12 and 89 m and within a temperature range from 16 to 25 °C. Abundance is maximal (up to 200 individuals per m2) at 19 °C, a temperature that also corresponds to the occurrence of the largest specimens (Tendal & Thiel, 2003). Although the bathymetric distribution of this and other schizamminid species may be influenced by sediment grain size (Buchanan, 1960), the need for a high food supply seems to the main factor controlling its overall range. Tendal & Thiel (2003) hypothesized that J. foetida is restricted to regions where seasonal upwelling occurs, which would be consistent with its large test size and likely high individual biomass. Other large agglutinated foraminifera are reported to occur in areas of organic matter flux to the seafloor (e.g., Gooday et al., 1992). The overall distribution coincides well with large parts of the Canary Current Upwelling System (CCUS), an area that extends from the Iberian Peninsula to Guinea, and constitutes one of the most productive coastal upwelling systems in the world (Demarcq & Somoue, 2015; Kämpf & Chapman, 2016). In addition, the CCUS area is situated adjacent to the Sahara Desert and exposed to one of the highest rates of airborne dust, a major source of nutrients, in particular iron (Neuer et al., 2004). Towards the southern part of the range (Gulf of Guinea), river runoff becomes the main source for the organic matter deposited on the continental shelf (Kämpf & Chapman, 2016). Despite extensive studies on the shallow benthic foraminiferal assemblages from reefs, shallow coastal habitats, lagoons and mangrove environments, J. foetida has not yet been recorded from Gabon, Sao Tomé, Príncipe or Nigeria (Langer, Fajemila & Mannl, 2016; Fajemila & Langer, 2016, 2017; Fajemila, Sariaslan & Langer, 2020).

Concluding remarks

Jullienella foetida is probably the largest agglutinated foraminiferal species occurring in relatively shallow water (<100 m depth). The thin, basically fan-shaped test can reach lengths of up to ~14 cm, a size only matched among continental-shelf foraminifera by the discoidal calcareous nummulitiid (Globothalamea) Cycloclypeus carpenteri Brady, 1881 (Briguglio et al., 2016). Some deep-sea xenophyophores are larger in terms of test size (up to 20 cm or more; Tendal, 1972), but only a small part (a few percent at most) of this volume is occupied by cytoplasm. In contrast, our new observations of the internal structure of J. foetida suggests that the cell body probably fills much of the test interior, which would mean that this species is possibly one of the largest of all foraminifera in terms of biomass. X-ray images of the test reveal an elaborate system of radial partitions that subdivides the test interior into channels (also shown in pl. 17, figs. 7, 8 of Loeblich & Tappan, 1987). These may serve to direct the flow of the cytoplasm, and perhaps increase its surface to volume ratio, as suggested recently for the much smaller calcareous foraminifera Chilostomella ovoidea Reuss, 1850 (Nomaki et al., 2020).

Jullienella foetida occupies a restricted geographical range around part of the NW African margin at water depths above 100 m (Fig. 1). It is found in eutrophic settings and on sandy, sometimes rippled substrates, suggesting a preference for energetic environments. Like some other large agglutinated foraminifera (Gooday et al., 1992), the cytoplasm contains diatoms, suggesting that it feeds on detritus. It seems likely that this species fulfils an important, perhaps keystone, ecosystem role by providing the only extensive firm substrate on which sessile organisms can settle (Cook, 1968, 1985; see also Figs. 7A, 7D), thereby increasing local biodiversity, as well as by processing organic matter at the base of the benthic food chain. However, much remains to be learnt about the ecology and biology of J. foetida. It will also be important to obtain DNA sequences from fresh material in order to clarify the place of this giant species, and others currently assigned to the Schizamminidae, within the radiation of monothalamous foraminifera.

Supplemental Information

Supplemental Information 1 Additional greyscale micro-CT images of the Jullienella specimen illustrated in Figures 2A, A’ and 4A and 4B with two density components.

(a, b, c, d) Sections in the plane of the test. The agglutinated test wall (=aw) and internal test partitions (=itp) are well defined as dense and bright white in greyscale scan images. The cytoplasm (=cy) occurs as low-density, material (light-grey, low-density component) and is patchily distributed throughout the test. MorphoSource ARK identifiers (a) ark:/87602/m4/393816; (b) ark:/87602/m4/393840; (c) ark:/87602/m4/393843; (d) ark:/87602/m4/393819.

Click here for additional data file.

We are grateful to Hjalmar Thiel (Hamburg) and Alexander Altenbach (Munich)† for providing additional information on the life position of Jullienella and to John Murray† (Southampton) for the collection of material from off Ghana. We also thank Georg Oleschinski and Martha Berens for assistances with the light microscopy, X-ray and SEM images and Rico Schellhorn for help with ImageJ. Sincere thanks are due to Antonino Briguglio, Willem Renema and an anonymous reviewer for constructive comments on the manuscript.

Additional Information and Declarations

Competing Interests

Author Contributions

Data Availability

The authors declare that they have no competing interests. Walid A. Makled is employed by the Egyptian Petroleum Research Institute (EPRI).

Martin R. Langer conceived and designed the experiments, performed the experiments, analyzed the data, prepared figures and/or tables, authored or reviewed drafts of the paper, and approved the final draft.

Anna E. Weinmann analyzed the data, prepared figures and/or tables, and approved the final draft.

Walid A. Makled analyzed the data, prepared figures and/or tables, and approved the final draft.

Janine Könen analyzed the data, prepared figures and/or tables, and approved the final draft.

Andrew J. Gooday conceived and designed the experiments, performed the experiments, analyzed the data, prepared figures and/or tables, authored or reviewed drafts of the paper, and approved the final draft.

The following information was supplied regarding data availability:

The raw CT scans and reconstructed 3D models are available at MorphoSource: Project 000393778.

https://www.morphosource.org/projects/000393778

-Fig 4A: https://doi.org/10.17602/M2/M393825

-Fig 4B: https://doi.org/10.17602/M2/M393822

-Fig 4C: https://doi.org/10.17602/M2/M393850

-Fig 4D: https://doi.org/10.17602/M2/M393854

-Fig 6A: https://doi.org/10.17602/M2/M393828

-Fig 6B: https://doi.org/10.17602/M2/M393831

-Fig S1A: https://doi.org/10.17602/M2/M393816

-Fig S1B: https://doi.org/10.17602/M2/M393840

-Fig S1C: https://doi.org/10.17602/M2/M393843

-Fig S1D: https://doi.org/10.17602/M2/M393819

-https://doi.org/10.17602/M2/M393790

-https://doi.org/10.17602/M2/M393797

-https://doi.org/10.17602/M2/M393800

-https://doi.org/10.17602/M2/M393803

-https://doi.org/10.17602/M2/M393806

-https://doi.org/10.17602/M2/M393809

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
