# Peer review of "New observations on test architecture and construction of Jullienella foetida Schlumberger, 1890, the largest shallow-water agglutinated foraminifer in modern oceans"

_PeerJ, doi:10.7717/peerj.12884_

## Round 0.1 · original submission · Minor Revisions

You provide a detailed study of the largest shallow-water agglutinated foraminifer known today using state-of-the-art methods allowing a better understanding of its morphology and taxonomic affiliation. I would like to see this important work published, but there are some minor but crucial issues which need to be addressed before publication. The main points that need to be addressed are:

1) Research question: please provide the main hypotheses/research questions to be addressed in the introduction (compare reviewer 1)

2) Material: please provide a more detailed description of the preservation of the specimens used for your study. Were they preserved in ethanol or just dried? (compare reviewer 1)

3) Computed tomography: Thank you for making the meshes and image series available through MorphoSource (when logging in with provided reviewer account details I was able to open them). Please make sure to add the identifiers (ark) to your manuscript at the latest upon publication. You mention a maximum resolution of 1 μm but the actual resolution seems more than a magnitude less. Please discuss how the comparable low resolution might affect your interpretations of the fine structure and the estimated biomass (compare Reviewers 1 and 3).

4) Test morphology and perforations: Reviewer 2 points out that test perforations are mentioned in the original description of the genus, while you rather refer to a thin veneer of angular agglutinated particles. Please clarify this discrepancy. You also mention the nature of the apertures, but these are not illustrated. I agree with reviewer 2 that I would be beneficial to figure the aperture in a non-broken specimen. I agree with reviewer 3 that it would also be beneficial to segment the inner parts of the test.

5) Interpretation of the extent of the cytoplasm: you assign Jullienella to the Schizamminidae where arborescent, agglutinated taxa where sediment particles stick in a thick organic lining (see reviewer 3). I agree with reviewer 3 that you need to explain why you interpret the grey matter rather as cytoplasm and not as organic lining. In addition, in poorly preserved specimens the denser material interpreted as cytoplasm could potentially even some sort of mold (see reviewer 1)

6) Computed tomography: Please address how the comparably low resolution might affect your interpretations.

Please make sure to address these points as well as all other point raised by the reviewers including those in annotated pdfs.

I look forward to receiving the revised manuscript.

·

Basic reporting

This is a well written, clear and interesting paper on description and illustration of possibly one of the largest living foraminifer.

Literature is adequate.

structure is sound and all seems to be well organized.

I found the material and method section lacking and important detail such as the state of preservation of the specimens. were they preserved in ethanol? where they just dried in a container. this is important information to check weather or not the less dense material included into the test is truly cytoplasm or just some sort of mold because they were poorly dried.

I could not really see and hypothesis and results to confirm it, this is just a descriptive papers that deals with accurate description of a poorly known taxon within the foraminifera.

Experimental design

there is not really a big research question behind this study. perhaps the need to fully understand the external and internal structure of this taxon is the highest goal of the authors. indeed this is an important piece of science, but nothing that derives from some primary research questions. As simple as that: we have a fancy CT scan and we have such poorly known foraminifera. let's see what is inside and let's see what we can find out to better describe such shape and form.

there is a quite consistent part of clear speculation regarding the biomass of this foraminifer. The authors are well aware of this and clearly state that the entire discussion is based on large assumptions (lines 254-257), but they still want to give an approximated value. Perhaps it can be done in less lines.

I think that there is a major concern regarding resoluzion of the CT scan and the eventual outcoming results. It seems to me that resolution of the scan is 16micrometers (see lines 92), which is consistent with such large specimens to scan. few lines below they state that the CT is actually capable of going down to 1 micrometer but I do not really get if this is the case. it seems to me not. if 16um is the voxel sizes of the scan, then the entire discussion on internal structure is very blurr and misleading. with 16um resolution a large variety of structures are simply not visible and recognizable. see more comments on this line directly in the MS.

I could not find the scan data uploaded to the given website. that needs to be checked.

Validity of the findings

the resolution issue needs to be resolved to clearly state that some structures cannot be visualized or directly do not exist.

conclusions also confirms already published data regarding structures and internal organization.

Reviewer 2 ·

Basic reporting

The manuscript by Langer et al. is a welcome contribution that elucidates the morphology and systematics of this larger agglutinated foraminifer that is abundant along the continental shelf of West Africa. I have always had my doubts about the systematic position of this genus, because it looks so much like a Xenophyophore. The current research proves that Jullienella does not belong in this group, and moreover provides a much improved description of the test morphology, wall structure, and cytoplasmic structure of the genus. The text of the paper is well-organised and well-written, and my comments and corrections are all very minor.

Experimental design

no comment

Validity of the findings

I only wish to make a few minor points: In the description of the genus given by Loeblich and Tappan (1987), the nature of the test perforations are mentioned: many large pores open into the interior but are constricted within the wall into smaller openings at the exterior”. The current authors note that the surface it the test is covered by a thin veneer of small angular agglutinated particles. The authors don’t mention any smaller openings, and no openings are visible in the SEM photos. Maybe Loeblich & Tappan had specimens that were abraded? The authors might with to clarify this point in the discussion (line 206).

Also, in the description of the test morphology, in line 125, the nature of the apertures are mentioned, but no illustrations of the natural apertures are given in the figures. Only broken specimens are illustrated in edge view. Do the authors have any images of the apertures that can be added to the figures?

Additional comments

My additional comments are as follows:

Line
359, 526. The publication date of the Loeblich & Tappan book is 1987, not 1988.
377. Sadly, John Murray passed away on October 22, 2021. The acknowledgement ought to be changed to “John Murray (Southampton)†”.
407. Please change “Wolfgring, S.” to “Wolfgring, E.”. The author’s name is Erik.

·

Basic reporting

THis a well written report on the occurrence of Julieniella.

Experimental design

Yes, the topic is in line with the aims and scope of the journal in my opinion. The aim of the paer is well defined, i.e. reporting the morphology of a curious foraminifera and its implications.

Validity of the findings

THe aouthors provide a very detailed description of the tests of the foraminifera Jullienella. This probably is one of the larger foraminifera currently alive. The manuscript is well written. My main comments are points that are not totally clear to me, or should be included in the test.
1) It is inferred that Jullienella is a Schizamminidae. I am familiar with the deep oceanic arborescent taxa in that family. These are also agglutinated, but also have a very thick organic lining of the test wall. The sediment particles appear to stick in/to this organic lining. In the text I miss in the discussion why the dark grey matter is interpreted as cytoplasm and how the possibility of it being an organic lining is excluded. Foram cell plasma is not dense at all, and in most CT-scans is only seen as a very small light blop when low voltages are used. These scans were produced at 120 kV and I would doubt that cytoplasma would be visible in the reconstructions.
Further on this, in most foraminifera the cytoplasm is not filling the entire cell cavity.
2) It would be nice to segment the inner parts of the test. This would be a good illustration of the shape and structure of the test cavity, or is this what is meant with the 'inner surface of the test wall in reverse'?

small points:
line 69 great => large
line 94 delete: and a maximum resolution (voxel size) of 1 um (irrerelevant what the max resolution is, and technically resolution and voxel size are not the same)
line 139: clarify 'in reverse view'
line 218: foraminfera => foraminifera (I am always happy to see that I am not the only one making this typo)

---

## Round 0.2 · accepted · Accept

Thank you for carefully addressing our suggestions. These small additions on preservation, cytoplasm, morphology and CT repository make the paper easier to follow and further widen its importance and scope. I look forward to seeing it published.